# Sesquiterpene Lactones as Promising Phytochemicals to Cease Metastatic Propagation of Cancer

**DOI:** 10.3390/biom15020268

**Published:** 2025-02-12

**Authors:** Fatemeh Mehdikhani, Homa Hajimehdipoor, Mojgan Tansaz, Marc Maresca, Sadegh Rajabi

**Affiliations:** 1Department of Clinical Biochemistry, School of Medicine, Shahid Beheshti University of Medical Sciences, Tehran 1985717411, Iran; fatemeh.mehdikhani1992@gmail.com; 2Department of Traditional Pharmacy, School of Traditional Medicine, Shahid Beheshti University of Medical Sciences, Tehran 1516745811, Iran; hajimehd@sbmu.ac.ir; 3Department of Traditional Medicine, School of Traditional Medicine, Shahid Beheshti University of Medical Sciences, Tehran 1516745811, Iran; tansaz_mojgan@sbmu.ac.ir; 4Aix Marseille University, CNRS, Centrale Med, ISM2, 13013 Marseille, France; 5Traditional Medicine and Materia Medica Research Center, Shahid Beheshti University of Medical Sciences, Tehran 1516745811, Iran

**Keywords:** sesquiterpene lactones, phytochemicals, cancer metastasis, signaling pathways, angiogenesis

## Abstract

Cancer metastasis remains the most challenging issue in cancer therapy. Recent reports show that cancer metastasis accounts for over 90% of cancer-associated deaths in the world. Metastasis is a multi-step process by which cancer cells spread to distant tissues and organs beyond the primary site. The metastatic propagation of different cancers is under the surveillance of several regulating processes and factors related to cellular signaling pathways. Plant-derived phytochemicals are bioactive components of plants with a variety of biological and medicinal activities. Several phytochemicals have been shown to target various molecular factors in cancer cells to tackle metastasis. Sesquiterpene lactones, as a diverse group of plant-derived phytochemicals with a variety of biological activities, have been shown to suppress the promotion and progression of different cancer types by acting on multiple cell-signaling pathways. This review article briefly describes the process of metastasis and its components. Then, sesquiterpene lactones with the ability to target and inhibit invasion, migration, and metastasis along with the molecular mechanisms of their effects on different cancers are described in detail.

## 1. Introduction

Cancer metastasis is a multi-step process that spreads cancer cells to distant tissues and organs beyond the primary site [1]. This accounts for over 90% of cancer-related deaths worldwide [2].

Metastasis plays a crucial role in the progression and prognosis of tumors [3]. It involves several stages: invasion, intravasation, survival in the bloodstream, extravasation, and colonization in secondary sites [4]. The tumor microenvironment (TME) plays a crucial role in the promotion and progression of metastasis. It contains various cell types, including immune cells, mesenchymal stem cells (MSCs), cancer-associated fibroblasts, adipocytes, endothelial progenitors, and mature cells [5]. Several secretory factors in the tumor microenvironment (TME), such as growth factors, cytokines, chemokines, hormones, metabolites, and extracellular matrix (ECM) components, play crucial roles in cancer promotion and metastasis. The metastatic secondary organ creates a favorable microenvironment that promotes the growth of a disseminated tumor [6]. The metastatic cancer cells undergo a process known as epithelial–mesenchymal transition (EMT) to have the ability to invade adjacent tissue [7]. EMT is a process by which epithelial cells gain a fibroblast-like phenotype by activating mesenchymal cell transcriptional events [7].

Phytochemicals are the natural bioactive ingredients of a variety of plants with beneficial health effects beyond basic nutrition [8]. They exhibit several desirable biological activities including anti-cancer, anti-inflammatory, anti-oxidant, and antimicrobial effects in vitro and in vivo [9]. Phytochemicals exert anti-cancer effects through different mechanisms. They induce cell death in cancer cells, target specific molecules in cellular pathways, modulate oxidative stress, and prevent tumor angiogenesis, which hinders metastasis [9]. Various phytochemicals have been shown to inhibit the metastatic propagation of cancer cells via several mechanisms. For example, curcumin is a polyphenol derived from Curcuma longa that can hamper cancer cell metastasis by inhibiting transcription factors, cell adhesion molecules, cell surface markers, and EMT [10]. Sesquiterpene lactones are a diverse group of plant-derived phytochemicals with anti-inflammatory, antiviral, antimicrobial, antimalarial, anti-cancer, antidiabetic, and analgesic properties [11]. Sesquiterpene lactones are derived from isopentenyl diphosphate and dimethylallyl diphosphate, key intermediates in the mevalonate and 2-C-methyl-D-erythritol pathways in the cytosol and chloroplast, respectively [12]. This review article briefly describes the process of metastasis and its components. Then, a list of sesquiterpene lactones that can target and inhibit invasion, migration, and metastasis processes along with underlying mechanisms in a variety of cancers will be provided.

## 2. The Process of Metastasis

One of the most critical hallmarks of cancer cells is their ability to metastasize. This process allows cancer cells to disseminate from their primary locations and spread to colonize various organs [13,14]. Cancer cells undergo a complex sequence of stages to spread and form secondary tumors in new locations. These complex processes are briefly described here. Figure 1 depicts different steps in the process of metastasis in cancer cells.

### 2.1. Invasion

To spread to other organs, tumor cells can migrate either individually as single cells or together as detached groups [15]. Certain morphological changes such as EMT in primary tumor cells and alterations in the ECM at the tumor site occur to help the tumor cells acquire invasive characteristics [10]. During the EMT, cancer cells change their phenotype from epithelial to mesenchymal [10]. This process is driven by various factors, including the hepatocyte growth factor (HGF), integrins, the platelet-derived growth factor (PDGF), fibroblast growth factor (FGF), and vascular endothelial growth factor (VEGF) [10]. Two key features of the EMT are the downregulation of E-cadherin, driven by transcriptional repressors such as Slug/SNAI2, SIP1/ZEB2, and Snail/SNAI1, and the upregulation of N-cadherin and vimentin. E-cadherin plays a crucial role in adherence junctions, and, within cells, it binds α-catenin, p120-catenin, and β-catenin, facilitating signaling and linking the actin cytoskeleton to the junctions. The overexpression of N-cadherin, on the other hand, leads to changes in the cytoskeletal structure. This switch from E-cadherin to N-cadherin enhances the motility of EMT-transformed cells. As a result, cells that undergo EMT lose their epithelial characteristics, enabling them to break away from epithelial cell clusters and migrate as individual cells in a mesenchymal manner [16].

### 2.2. Angiogenesis

For tumor cells to grow and survive, providing nutrients and oxygen is essential. The tumor cells obtain them by positioning themselves close to blood vessels for access to the circulatory system [17]. Cancer cells create vessels via two main methods: forming new blood vessels or leveraging existing ones [17,18]. During angiogenesis, pro-angiogenic signals activate, prompting new blood vessel formation and allowing tumor cells to grow rapidly [19]. Tumor-associated inflammation, immune cell involvement, genetic changes in malignant cells, hypoxia, and pro-angiogenic factor expression are critical for the angiogenic switch [17]. Under low-oxygen conditions, factors such as VEGF and FGF are produced to stimulate angiogenesis [20].

### 2.3. Intravasation

The infiltration of malignant cells into lymphatic or blood vessels is referred to as intravasation [21]. This step is crucial in the metastasis process, as it enables tumor cells to spread and colonize secondary sites. Tumor cells can more easily intravasate into lymphatic vessels compared to blood vessels due to the looser inter-endothelial junctions [22,23]. Intravasation occurs in two phases: first, malignant cells are guided toward the vessel wall by chemotactic signals from the tumor microenvironment; then, they penetrate the vessels [21]. During intravasation, cancer cells can migrate as single cells using an amoeboid invasion mechanism, allowing them to move through areas lacking ECM, or through a mesenchymal invasion mechanism, where they enzymatically degrade dense ECM fibers [24,25]. Tumor cells also express VEGF-A extensively, which increases vessel permeability, facilitating their entry [26]. Additionally, tumor-associated macrophages play a significant role in guiding tumor cells toward blood vessels [27].

### 2.4. Circulation and Extravasation

When tumor cells enter the bloodstream, becoming circulating tumor cells (CTCs), they encounter challenging conditions [1]. To ensure survival and reach distant organs, CTCs interact with various components in the circulatory microenvironment [28,29]. Research indicates that the circulating clusters of tumor cells have a higher metastatic potential compared to single cells [30]. These clusters often contain immune and stromal cells, which enhance their survival [30,31,32]. For example, CTCs can engage with platelets, forming a protective coating that shields them from immune cell detection [33,34]. Additionally, the presence of neutrophils within these clusters aids in their survival by suppressing leukocyte activity [35]. CTCs can attach to and extravasate through endothelial cells, allowing them to colonize the pre-metastatic niche (PMN) [21]. Upon reaching capillaries, they may either continue to move within the vessels before ultimately extravasating and colonizing the PMN, or they may extravasate directly through endothelial migration [36,37,38]. Paracellular migration is the most common form of extravasation, where cancer cells pass between two endothelial cells. This process involves the breakdown of inter-endothelial cell junctions and the cellular reorganization of the surrounding cells [36,39].

### 2.5. Colonization

Among CTCs, only a small fraction successfully reaches secondary sites, where they may either develop into macrometastatic tumors or enter a dormant state. The decision between growth and dormancy is influenced by various extracellular and intracellular signals, as well as factors within the bone marrow niche [1]. For instance, evidence suggests that actin assembly plays a role in shifting cells from dormancy to active growth [40]. Additionally, the balance between p38 and ERK signaling pathways is critical: a higher ERK/p38 ratio favors dormancy, while a higher p38/ERK ratio promotes progression [1]. Furthermore, research indicates that tumor cells selectively metastasize to and colonize specific organs. For example, tumor-derived exosomes carrying hyaluronan-binding protein (CEMIP) and cell migration-inducing factors show a particular affinity for the brain over other organs [41]. This phenomenon is known as organotropism, where different tumors exhibit preferred metastatic sites based on the origin of the primary tumor [6].

## 3. Sesquiterpene Lactones with the Ability to Inhibit Metastasis

Sesquiterpene lactones represent a diverse and extensive group of natural compounds found in over 100 families of flowering plants, typically extracted from their aerial parts or leaves [42]. Numerous studies have explored the pharmacological properties of sesquiterpene lactones, revealing their anti-tumor, antiviral, antimicrobial, and anti-inflammatory effects [43]. These phytochemicals attach to the free sulfhydryl groups of proteins, disrupting their functions and influencing various biological processes such as cell growth, signaling pathways, cellular respiration, and apoptosis [44]. Notably, sesquiterpene lactones exhibit significant anti-cancer properties, particularly in inhibiting metastasis. In vitro studies indicate that sesquiterpene lactones can reduce the expression of cell adhesion molecules (ICMs), which are essential for the spread and migration of malignant cells [45]. Figure 2 illustrates the chemical structure of sesquiterpene lactones that have anti-metastatic properties.

### 3.1. Alantolactone

Alantolactone (ALT), a type of sesquiterpene lactone isolated from *Inula helenium,* shows potential as an anti-cancer agent for various cancer types [46]. In a study conducted by Liu et al., ALT was found to inhibit angiogenesis through both in vitro and in vivo experiments. In vitro, using human umbilical vascular endothelial cells (HUVECs), ALT significantly reduced cell mobility and migration. In vivo, the use of the chick embryo chorioallantoic membrane (CAM) assay demonstrated that ALT suppressed new blood vessel formation. Additionally, in a study using MDA-MB-231 xenografts in mice, ALT treatment resulted in reduced tumor size and weight, likely due to the inhibition of angiogenesis. The study also revealed that ALT inhibited the VEGF-triggered activation of VEGFR2 phosphorylation in HUVECs, leading to the downregulation of VEGFR2 signaling pathways, including FAK, Src, Akt, and PLCγ1 [46]. In another study, the effects of ALT on the progression and migration of MCF-7 cells were examined. That investigation demonstrated that ALT significantly suppressed colony formation and migration in these cells. Additionally, the study measured the levels of matrix metalloproteinases (MMPs), a key family of proteases involved in cell migration during metastasis, known for their ability to degrade the extracellular matrix (ECM) and basement membrane. The results also showed that ALT inhibited invasion and cell migration by downregulating MMP-2, MMP-7, and MMP-9 [47].

One of the challenges in treating melanoma patients with mitogen-activated protein kinase inhibitors (MAPKis), which target MAPK signaling pathways, is the development of drug resistance due to the activation of the signal transducer and activator of the transcription 3 (STAT3) signaling pathway. STAT3 activation promotes melanoma cell proliferation, invasiveness, and metastasis. To address this issue, a study suggested that combining MAPKi with ALT might offer a promising treatment strategy. That study demonstrated that ALT can effectively inhibit STAT3 signaling, potentially overcoming the resistance and improving therapeutic outcomes [48]. The anti-metastatic effects of ALT were also evaluated in prostate cancer cells exhibiting stem-like characteristics. The study revealed that ALT antagonized the STAT3 signaling pathway, leading to the upregulation of p53 and the downregulation of Oct-4, CD44, CD133, and Nanog expression. Consequently, the research suggested that ALT can reduce stemness traits and inhibit migration in metastatic prostate cancer cells [49]. Osteosarcoma is a common and lethal form of bone cancer.

The PI3K/AKT signaling pathway plays a crucial role in promoting tumorigenesis, aggressiveness, and metastasis in cancer cells, making it a potential therapeutic target, particularly in metastatic osteosarcoma. In a study conducted by Zhang et al., it was found that ALT effectively inhibited the PI3K/AKT signaling pathway in U2OS and HOS osteosarcoma cell lines, thereby reducing cell migration, invasion, and aggressiveness [50]. Another study investigated the effects of ALT on 143B, MG63, U2OS, and SaoS2 human osteosarcoma cell lines and found a decrease in EMT-related markers, including vimentin, Snail, and N-cadherin, while the epithelial marker E-cadherin increased. Additionally, cell invasion, migration, and proliferation were reduced. ALT treatment also led to a decrease in the expression of MMP-9, MMP-2, and MMP-7, suggesting its potential to inhibit metastasis. Furthermore, the study revealed that ALT negatively affects the Wnt/β-catenin and MAPK signaling pathways, both of which are abnormally activated in metastatic osteosarcoma. The combination of ALT with Wnt/β-catenin and MAPK inhibitors further suppressed osteosarcoma growth, aggressiveness, and metastasis [51]. Aldo-keto reductase family 1 member C1 (AKR1C1) is shown to be involved in the metastasis of cancer cells, particularly in non-small-cell lung cancer (NSCLC). The treatment of NCI-H460 cell lines and subcutaneous NCI-H460 cell xenograft tumors with ALT demonstrated that ALT could inhibit AKR1C1 by binding to it. This inhibition resulted in a reduction in AKR1C1 expression, as well as decreased metastasis and cell growth [52]. These studies collectively suggest that ALT may have significant potential as a candidate for halting metastasis.

### 3.2. Ambrosin

Ambrosin is a pseudoguaianolide sesquiterpene that can be derived from ragweed species [53]. Research indicates that ambrosin exhibits a range of therapeutic effects, including anti-tumor activity [54]. A study involving human breast cancer cell lines MCF-7, JIMT-1, and HCC1937, as well as the normal-like breast epithelial cell line MCF-10A, demonstrated that treatment with 5 μM ambrosin reduced the populations of breast cancer stem cells. Additionally, in the JIMT-1 cell line, ambrosin treatment inhibited cell migration, suggesting its anti-metastatic activity against breast cancer [55]. Another study evaluated the impact of ambrosin on the triple-negative MDA-MB 231 cell line and found that it inhibited cell proliferation and led to the downregulation of the Wnt/β-catenin pathway [56]. In another study, the effects of ambrosin on MDA-MB 231 cells were investigated. The findings revealed that ambrosin inhibited colony formation and reduced MMP expression in these highly metastatic breast cancer cells. Additionally, the study unraveled that ambrosin decreased the levels of phosphorylated GSK-3β and Akt, thereby inhibiting the Akt/β-catenin signaling pathway [57].

### 3.3. Antrocin

Antrocin, a natural compound derived from *Antrodia camphorata*, has shown therapeutic potential in cancer treatment [58]. To uncover the molecular mechanisms of antrocin’s effects on cancer, Chiu et al. conducted a study using human bladder cancer cell lines (5637 and T24) treated with antrocin. Their findings revealed that antrocin effectively inhibited invasion, migration, and cell proliferation. Focal adhesion kinase (FAK), a non-receptor tyrosine kinase, plays a key role in regulating cellular motility and morphology. FAK relays signals from growth factor receptors or integrins to Src, PI3K, p130Cas, paxillin, and Grb7, which in turn activate the Rho/Ras/Cdc42 or MAPK pathways, thereby influencing motility and cellular structure. In tumors, FAK expression is significantly elevated. In this study, antrocin notably reduced the phosphorylation of FAK and paxillin, leading to the disruption of filopodia and lamellipodia formation due to altered FAK and paxillin distribution. Additionally, antrocin increased E-cadherin levels, decreased vimentin expression, and reduced MMP-2 activity [59]. Both in vitro and in vivo, treating breast cancer cells with antrocin led to a reduction in migration, tumorigenesis, and proliferation. This suppression was achieved by downregulating the expression of oncogenes and stemness-related markers such as β-catenin, Akt, and Notch1 [60]. Furthermore, a study conducted on human renal cell carcinoma RCC 786-0 cells demonstrated that antrocin inhibited the Src, FAK, and ERK1/2 signaling pathways. This inhibition led to a reduction in the phosphorylation of paxillin, C/EBP-β, and total c-Fos levels. Additionally, the expression of MMP-7 and vimentin was decreased. As a result, antrocin treatment effectively disrupted cell migration, invasion, and the formation of lamellipodia [61].

### 3.4. Artemisinin

Artemisinin (ART) is a sesquiterpene compound extracted from the Chinese plant *Artemisia annua* with potential anti-cancer effects [62,63]. In a study involving human lung cancer A549 and H1299 cell lines, treatment with ART significantly inhibited the migration and invasion of these cancer cells as well as the activity of MMPs in a concentration-dependent manner. ART treatment also suppressed the expression of EMT-related proteins, including N-cadherin and vimentin, as well as cancer stem cell (CSC) markers like Nanog, Sox2, and Oct3/4, while increasing E-cadherin expression. Furthermore, ART disrupted the Wnt/β-catenin signaling pathway [64]. In experiments with human hepatocellular carcinoma (HCC) cell lines HepG2 and SMMC-7721, treatment with varying concentrations of ART resulted in a decreased expression of MMP2 and an upregulation of TIMP2, which is an inhibitor of MMP2. Additionally, ART treatment inhibited the activation of p38 and ERK1/2. It also enhanced cell adhesion by increasing Cdc42 activity, which activated E-cadherin. Collectively, these in vitro and in vivo findings demonstrate that ART treatment effectively inhibits cell motility, migration, and metastasis [65]. A study by Rasheed et al. demonstrated that ART treatment in non-small-cell lung cancer (NSCLC) cell lines inhibited the expression of MMP-2, MMP-7, and u-PA, leading to the inhibition of metastasis and invasion. Furthermore, the CAM assay in that study revealed that ART treatment significantly decreased the number of metastasized cells and reduced tumor size [66].

### 3.5. Brevilin A

Another sesquiterpene lactone is brevilin A (BA), which can be extracted from *Centipeda minima* and may have potential applications in cancer therapy [67,68]. A study evaluated the effects of BA on melanoma cancer cell lines A375 and A2058, as well as in a mouse A375 xenograft model of this cancer. The findings demonstrated that BA inhibited the JAK2/STAT3 pathway by reducing the phosphorylation of JAK2 and STAT3. According to the results of that study, BA treatment effectively suppressed cell invasion and migration [69]. In a study involving HCC cell lines, including HepG2 and SMMC-7221, treatment with BA led to the downregulation of MMP-2 and MMP-9. Additionally, BA inhibited the Wnt/β-catenin and STAT3/Snail signaling pathways, resulting in decreased cell invasion [70]. The effects of BA were evaluated on a xenograft mouse model of breast cancer and two triple-negative breast cancer (TNBC) cell lines, MDA-MB 231 and MDA-MB 468. The study found that BA inhibited cell migration and reduced the phosphorylation and expression of Akt, mTOR, and STAT3, thereby suppressing the Akt-mTOR and STAT3 signaling pathways. In the mouse xenograft model, tumor growth was significantly reduced [71]. In an in vitro study using HCT-116 and CT26 colorectal cancer (CRC) cell lines, BA was found to negatively impact the expression of MMP-2 and VEGF and inhibit STAT3 activation. The research also demonstrated that BA suppressed angiogenesis and reduced cell migration and invasion [72]. In another study, CRC cells were co-cultured with hepatic stellate cells (HSCs), and an in vivo experiment was performed to evaluate the effects of BA. The findings revealed that BA effectively inhibited colorectal liver metastasis and tumor growth by targeting the VEGF-IL6-STAT3 axis [73].

### 3.6. Bigelovin

*Inula helianthus-aquatica* C. Y. Wu is one of the plants traditionally used for the treatment of various complications. Research indicates that bigelovin, a sesquiterpene lactone derived from this plant, possess potential to be used for cancer treatment [74]. A study using THP-1 human monocytic and HMEC-1 human microvascular endothelial cell lines along with zebrafish embryos demonstrated that bigelovin suppressed the formation of subintestinal vessels in zebrafish embryos. Additionally, it downregulated the expression of angiogenesis-related genes such as *Ang-1*, *Ang-2*, *Tie-1*, and *Tie-2*. The study also found that Th1 cytokines, including IFN-γ, IL-2, and IL-12, were not produced. Furthermore, bigelovin inhibited the expression of CAM genes like *ICAM-1*, *VCAM-1*, and *E-selectin*, which are associated with inflammation, and prevented human monocytes from adhering to human endothelial cells. These findings indicate that bigelovin has an anti-angiogenic effect [75]. The anti-metastatic properties of bigelovin were demonstrated in a study on murine colon cancer cells (colon 26-M01) and human colon cancer cells (HCT116). The treatment led to significant changes in key molecules, including p-STAT3, STAT3, Rock, β-catenin, N-cadherin, Rac1/2/3, and RhoA, resulting in the disruption of the IL6-STAT3 and cofilin pathways. This disruption inhibited cell motility, migration, EMT, angiogenesis, and cell growth. Additionally, in orthotopic colon tumor allograft-bearing mice, bigelovin modulated the tumor microenvironment by increasing macrophages and T lymphocytes, ultimately suppressing liver and lung metastasis [76].

### 3.7. Britannin

Britannin (BRT) is a sesquiterpene lactone compound that can be extracted from *Inula britannica* L and is considered a potential anti-cancer agent [77]. Studies suggest that the transcription factor Twist-related protein 1 (TWIST1) plays a significant role in EMT and is crucial for metastasis. Additionally, COX-2 is associated with tumor invasion and angiogenesis. The treatment of human gastric cancer cell lines, such as AGS and MKN-45, with BRT, resulted in a reduction in the expression of MMP-9, TWIST-1, and COX-2. These findings suggest that BRT has potential as a drug to prevent tumor metastasis [78]. An investigation involving HCC cell lines, including BEL-7402 and HepG2, demonstrated that treatment with BRT suppressed tumor cell migration [79]. Studies have indicated that the Programmed Death-Ligand 1 (PD-L1) protein is associated with angiogenesis in cancer. In one study, treating human umbilical vein endothelial cells (HUVECs) with BRT inhibited invasion, migration, and angiogenesis by reducing PD-L1 levels. Additionally, in human colon cancer HCT116 cells treated with BRT, the expression of VEGF and MMP-9 was suppressed. This research suggested that BRT can inhibit metastasis and angiogenesis by targeting PD-L1 [80]. Kruppel-like factor 5 (KLF5) is a transcription factor crucial for cell growth and survival. In various tumors, KLF5 is often overexpressed. In an in vitro study, the treatment of A549 lung carcinoma cells with BRT reduced KLF5 expression and inhibited cell migration [81].

### 3.8. Costunolide

Costunolide (CE), the primary compound found in *Saussurea lappa*, exhibits anti-tumor properties [82]. In a study conducted by Choi et al., using MDA-MB 231 breast cancer cell lines, CE treatment inhibited NF-*κ*B, which led to the suppression of TNF*α*-induced migration and the invasion of these cells. Furthermore, NF-*κ*B inhibition resulted in the downregulation of MMP-9. Additionally, CE effectively hindered metastasis in an orthotopic mouse model of MDA-MB 231 breast cancer [83]. The effects of CE on colorectal cancer cell lines HCT-15, HCT-116, and DLD1 were assessed in a study, revealing a significant reduction in cell migration and invasion compared to control cells. A Western blot analysis of EMT markers also showed that CE treatment resulted in a decrease in vimentin and N-cadherin levels, along with an increase in E-cadherin expression. These findings suggest that CE treatment effectively inhibits metastasis [84]. The anti-angiogenic effects of CE were evaluated through both in vitro and in vivo studies, involving human umbilical vein endothelial cells (HUVECs), mouse models of VEGF-induced neovascularization, and KB3-1 human epidermoid carcinoma cells. The results demonstrated that CE interfered with the VEGFR KDR/Flk-1 signaling pathways associated with angiogenic factors, leading to the suppression of pro-angiogenic activity [85]. In a study on the human non-small-cell lung cancer cell line H1299 treated with CE, wound healing and transwell assays demonstrated that CE inhibited cell migration in a concentration-dependent manner and invasion in both time- and concentration-dependent manners. Additionally, markers related to EMT were evaluated, revealing an upregulation of E-cadherin and a downregulation of N-cadherin. This evidence suggests that CE can suppress the EMT process in H1299 cells. Furthermore, CE treatment led to a reduction in the mRNA expression levels of integrins α2 and β1, as well as MMP2, indicating its role in metastasis inhibition [86]. In neuroblastoma NB-39 cell lines treated with CE, a significant inhibition of cell migration and invasion was observed. Additionally, CE treatment led to the downregulation of MMP-2, proposing that the reduced expression of MMP-2 may be linked to the suppression of cell invasion and migration [87].

### 3.9. Cynaropicrin

Cynaropicrin is a sesquiterpene lactone separated from *Cynara scolymus* L. [88]. Zheng et al. conducted a study on colorectal cancer cell lines HCT116, RKO, and DLD-1 by treating them with cynaropicrin. In this study, the results of transwell assays demonstrated a significant reduction in cell migration [89]. In another study, human melanoma A375 cells were treated with cynaropicrin. The wound healing assay and Boyden chamber invasion assay demonstrated the inhibition of cell motility and invasion. Additionally, the colony formation ability of A375 cells decreased following cynaropicrin treatment. Furthermore, the abnormal activation of the MAPK pathway is linked to cell motility, proliferation, invasion, and survival. ERK, located downstream of the MAPK signaling pathway, is hyperactivated in melanoma cells with BRAF mutations. This study showed that cynaropicrin treatment led to a reduction in both the MAPK/ERK and NF-κB pathways [90].

### 3.10. Dehydrocostus Lactone

Dehydrocostus lactone (DHC) is a sesquiterpene lactone isolated from *Aucklandiae Radix,* with known anti-tumor properties [82]. To validate its anti-tumor effects, Su et al. treated non-small-cell lung cancer (NSCLC) H1299 cells with DHC. In that study, wound healing and transwell assays were performed to demonstrate that DHC can significantly inhibit cell migration and invasion. Additionally, RT-qPCR analysis revealed that the expression of E-cadherin was upregulated, while N-cadherin, Snail, integrin α2, and MMP-2 were downregulated. These findings confirm the anti-metastatic activity of DHC against lung cancer [91]. The inhibitory role of DHC was further demonstrated in another study, where laryngeal carcinoma cells (TU212 and HBE) were unable to migrate or invade in the presence of DHC, which was attributed to the downregulation of MMP-2 and MMP-9 [92].

### 3.11. Deoxyelephantopin

Deoxyelephantopin (DOE) is a compound found in the traditional medicinal plant *Elephantopus scaber*. Studies have shown that DOE has the potential for cancer treatment [93]. For example, the anti-metastatic effects of DOE were evaluated on various cancer cell lines, including HCT 116 (colorectal), K562 (chronic myeloid leukemia), KB (oral), and T47D (breast) cell lines. This sesquiterpene lactone suppressed cell migration and invasion in these cancer cell lines. The activation of MMPs relies on the binding of uPA to its receptor uPAR, which converts plasminogen into plasmin. MMPs can also be inhibited by the tissue inhibitors of matrix metalloproteases (TIMPs), preventing ECM degradation. In a study by Kabeer et al., DOE treatment decreased the expression of uPA, uPAR, MMP-2, and MMP-9, while upregulating TIMP-1 and TIMP-2 [94]. Moreover, another study demonstrated the inhibition of metastasis by DOE in A549 cells, where migration and invasion were significantly reduced. The expression levels of NF-kB, IkBa, MMP-2, MMP-9, uPA, and uPAR were decreased, while TIMP-2 levels increased. Western blotting results also showed reduced protein levels of p-ERK 1/2 and p-Akt, along with increased levels of p-p38 and p-JNK, contributing to metastasis suppression [95].

In the study by Cvetanova et al., the migratory ability of A375LM5IF4g/Luc (a lung-seeking metastatic melanoma cell line) treated with DOE was evaluated. Key metastasis-associated markers, including N-cadherin, MMP2, vimentin, and integrin-4 were found to be inhibited. Additionally, when this cell line was injected into NOD/SCID mice, pulmonary vascular permeability, the angiogenesis marker VEGF, the neovascularization marker CD31, and N-cadherin in the tumor microenvironment of the lungs were suppressed. This led to the inhibition of lung metastasis due to the presence of DOE [96]. In a murine mammary adenocarcinoma cell line (TS/A cells), the anti-metastatic activity of DOE was compared with Paclitaxel (PTX), a diterpene alkaloid originally derived from *Taxus brevifolia*. The results demonstrated that DOE more significantly inhibited cell motility than PTX. Since cell motility can be regulated by calpain, DOE likely disrupted adhesion formation by inhibiting m-calpain’s enzymatic activity. Additionally, DOE treatment inhibited lamellipodia formation and actin filament organization [97]. In an in vitro and in vivo study conducted by Huang et al., combinational therapy using cisplatin and DEO was shown to inhibit cell migration in the B16 murine melanoma cell line and suppress lung metastasis in a murine B16COX-Luc metastatic allograft model [98].

### 3.12. Eupalinolide

*Eupatorium lindleyanum* DC. is a traditional Chinese medicinal herb with key SL components such as eupalinolide J, eupalinolide B, eupalinolide I, and eupalinolide G, with potential anti-tumor properties [99,100]. For instance, the treatment of highly metastatic MDA-MB-468 breast cancer cells with eupalinolide J demonstrated its STAT3 inhibitory effects. Additionally, the viability of both MDA-MB-468 and MDA-MB-231 cells was significantly suppressed [101]. The treatment of pancreatic cancer cell lines, including MiaPaCa-2, PANC-1, and PL-45, with eupalinolide B revealed a significant inhibition of cell migration and invasion. These findings highlighted the anti-metastatic properties of eupalinolide B [102]. In another study, the inhibitory effects of Eupalinolide J on MDA-MB-231, MDA-MB-468, and MCF-7 cell lines were evaluated. The results revealed a significant downregulation of STAT3 and p-STAT3 [100]. In a study conducted by Hu et al., U251 and MDA-MB-231 cells were treated with Eupalinolide J for 12 h, and then, the protein and gene levels of metastasis-associated factors were measured. The results revealed a significant decrease in STAT3, p-STAT3, MMP-9, and MMP-2, while PI3K and AKT remained unaffected. These findings indicated that Eupalinolide J degrades STAT3, leading to the inhibition of metastasis. Additionally, in a lung metastasis model using BALB/c nu/nu female mice, treatment with Eupalinolide J was found to suppress cancer cell metastasis [103].

In a study on human HCC cell lines, including MHCC97-L and HCCLM3, eupalinolide A treatment suppressed cell motility and migration through the downregulation of vimentin and the upregulation of ZEB1, N-cadherin, and fibronectin. These findings demonstrated that eupalinolide A can reverse the EMT process [104]. Additionally, Jiang et al. demonstrated that eupalinolide B treatment in laryngeal cancer TU212 cells inhibited EMT markers evidenced by a decrease in N-cadherin expression and an increase in E-cadherin. Migration assays also revealed a significant reduction in cell motility. Notably, the inhibition of lysine-specific demethylase 1 (LSD1) has been shown to prevent tumor cell migration and growth while inducing apoptosis. In that study, eupalinolide B effectively suppressed LSD1 [105].

### 3.13. Gaillardin

Gaillardin is a type of SL with anti-cancer properties, extracted from the aerial parts of *Inula oculus-christi* [106]. To demonstrate the anti-metastatic effects of gaillardin on tumor cells, Roozbehani et al. conducted a study on human gastric adenocarcinoma cell lines MKN45 and AGS. Their findings revealed that gaillardin treatment led to the suppression of NF-κB, which subsequently downregulated its target genes, including COX-2, TWIST-1, and MMP-9. These genes are associated with angiogenesis, cell invasion, and metastasis [107].

### 3.14. Helenalin

Helenalin is another SL derived from plants such as *Arnica montana* and *Arnica chamissonis ssp*. *Foliosa* [108]. This natural compound has been shown to exhibit anti-tumor properties. In a study on rhabdomyosarcoma (RMS) cell lines, including RH30 and RD, helenalin treatment significantly inhibited cell migration, as demonstrated by wound healing assays [109].

### 3.15. Isoalantolactone

Isoalantolactone is an SL derived from *Inula helenium* L., known for its anti-tumor activities [110]. In a study on the liver cancer cell line (Hep-G2), treatment with isoalantolactone significantly reduced cell invasion and migration, as shown by the transwell assay [111]. Additionally, in another study on MDA-MB-231 cell lines, isoalantolactone treatment inhibited the p38 and MAPK/NF-κB signaling pathways, leading to the suppression of cell migration and invasive activities [112]. In another investigation, a compound mixture called F35, composed of alloalantolactone, alantolactone, and isoalantolactone, was used to treat pancreatic cancer cell lines PANC-1 and SW1990. The treatment significantly suppressed colony formation and cell migration [110]. Moreover, isoalantolactone treatment in colorectal cancer cell lines HCT116 and SW620 significantly suppressed colony formation [113]. In endometrial cancer cell line HEC-1-B, isoalantolactone treatment inhibited both the migratory properties and invasiveness of the cell line [114]. The transwell assay on the liver cancer cell line HuH7 treated with isoalantolactone revealed significant inhibition of cell invasion [115]. The treatment of human pancreatic ductal adenocarcinoma (PDAC) cell lines, including PANC-1, AsPC-1, and BxPC-3, with isoalantolactone, led to a significant arrest in cell migration in PANC-1 cells and the inhibition of invasion in all three cell lines [116].

### 3.16. Isodeoxyelephantopin

*Elephantopus scaber* L. is a traditional Chinese medicinal herb with many biological activities. Isodeoxyelephantopin (ESI), a sesquiterpene lactone derived from this plant, has demonstrated anti-tumor activity in various types of cancer [117]. The effects of ESI on lung cancer cells were evaluated in a study conducted by Wang et al. The results of that study showed that ESI suppressed colony formation in lung cancer cell lines, including H1299 and A549. However, it did not inhibit the colony formation ability of non-cancerous HBE lung epithelial cells [118]. In a study on the MDA-MB-231 breast cancer cell line, the scratch wound migration assay proved that ESI may prevent cell migration. Additionally, real-time PCR analysis revealed that ESI treatment inhibited the expression of MMP-2 and MMP-9 [119]. In an in vitro study using H1299 lung adenocarcinoma cells, ESI inhibited cell invasion. Western blot analysis further demonstrated that ESI suppressed the expression of MMP-9 and ICAM-1 [120].

### 3.17. Lactucopicrin

Lactucopicrin is an SL found in various species of the Asteraceae family such as *Lactuca virosa*. This compound also exhibits anti-tumor properties [121]. Among various cancer types, the activation of the mTOR/PI3K/AKT signaling pathway can contribute to tumor progression. In the SKMEL-5 skin cancer cell line, treatment with lactucopicrin led to a reduction in p-PI3K, p-Akt, and p-mTOR levels in a dose-dependent manner [122]. Osteosarcoma cells have the ability to metastasize to distant organs. A transwell assay demonstrated that treating Saos-2 osteosarcoma cells with lactucopicrin inhibited cell migration and invasion in a dose-dependent manner [123]. Rotondo et al. treated U87MG glioblastoma cells with lactucopicrin and observed a significant reduction in colony formation. Additionally, the wound healing assay demonstrated decreased cell motility, and Western blot analysis revealed a reduction in Akt phosphorylation levels [124].

### 3.18. Parthenolide

There is evidence suggesting that parthenolide (PTL) has anti-metastatic properties [125]. This SL is derived from the plant *Tanacetum parthenium* [126]. A study evaluating the effects of PTL on renal cell carcinoma revealed a significant decrease in cell proliferation in 786-O and ACHN cells, as demonstrated by colony formation assays. Transwell assays further showed that PTL inhibited both cell migration and invasion. Additionally, markers associated with EMT and metastasis were assessed, revealing that MMP-2 and MMP-9 expression levels were significantly suppressed, while E-cadherin levels increased, and N-cadherin, vimentin, and Snail decreased. Markers related to stemness, including ALDH1, CD133, Oct4, and Sox2, were also evaluated using Western blot analysis, showing that these markers were inhibited by PTL treatment. Mammosphere formation assays, which assess cancer stem cell characteristics, indicated a significant reduction in the number of spheres. Furthermore, PTL treatment led to a decline in p-PI3K and p-AKT expression, suggesting the suppression of the PI3K/AKT pathway [125]. In another study, Jafari et al. investigated the anti-cancer activity of PTL. Real-time PCR analysis revealed that PTL downregulated vimentin expression in MCF-7 breast cancer cell lines [127]. The EMT pathway can be induced by HIF-1. In an in vitro and in vivo study, the effects of PTL under hypoxic conditions were compared in human colorectal cancer (CRC) cells. The results showed that the treatment of CRC cell lines (HT-29, DLD-1, and HCT116) with various concentrations of PTL significantly decreased HIF-1α expression. Using a matrigel-coated transwell chamber and wound healing assays, it was observed that cell migration and invasion increased under hypoxia, while PTL treatment mitigated these effects. Hypoxia is known to enhance MMP activity; however, PTL treatment reduced the expression of MMP-2 and MMP-9. Additionally, PTL increased E-cadherin levels compared to hypoxic conditions, while EMT markers such as β-catenin, vimentin, Slug, Snail, and Twist were downregulated. To confirm the in vitro findings, an in vivo experiment was performed using a nude mice model of CRC. CRC cells injected into the mice and treated with PTL showed reduced levels of CA IX, a marker of hypoxia. Immunohistochemistry revealed that PTL treatment decreased the number of NF-κB subunit p65-positive cells. Moreover, angiogenesis marker von Willebrand factor (VWF) and EMT marker vimentin were significantly reduced. These findings suggest that PTL inhibits the NF-κB/HIF-1α/EMT pathway in CRC cells, suppressing migration, invasion, and angiogenesis [128].

In another study on CRC cell line SW620, PTL treatment inhibited cell migration and invasion in a concentration-dependent manner. PTL treatment also led to the upregulation of E-cadherin and the downregulation of β-catenin, Snail, vimentin, MMP-2, MMP-9, and COX-2 expression [129]. In another study using both in vitro and in vivo models of LM8 osteosarcoma cells, PTL treatment resulted in a dose-dependent suppression of NF-κB DNA binding and transcriptional activity, as well as a reduction in VEGF expression. The matrigel invasion assay demonstrated that tumor invasion was significantly inhibited by PTL. In an LM8 murine model of osteosarcoma, PTL treatment considerably reduced pulmonary metastasis. IHC results showed that p65 expression was decreased in lung metastatic nodules of PTL-treated mice compared to untreated controls. Additionally, IHC revealed that VEGF expression in metastatic lung tumors and surrounding lung tissue was significantly suppressed in PTL-treated models. To further evaluate PTL’s effect on the different stages of lung metastasis, tumor cells were injected into the mice’s tail veins. The data showed that PTL suppressed the early stages of lung colonization but did not affect other steps before extravasation. This indicates that PTL likely inhibits the metastatic process by targeting critical early events in lung colonization, particularly within the first 48 h after tumor cells enter the lungs [130].

In a study on the MDA-MB 231 breast cancer cell line, treatment with PTL and its soluble analog dimethylamino parthenolide (DMAPT) led to the downregulation of NF-κB activity. Additionally, DMAPT treatment inhibited cell migration and reduced vimentin expression. In addition, in xenograft model of MDA-MB 231 in nude mice, DMAPT treatment resulted in a significant reduction in the levels of VEGF, MMP2, MMP9, and p65 [131]. In tumor cells, TGF-β1 promotes EMT and cancer stem cell characteristics. A study on CRC cell lines assessed the effects of TGF-β1 and PTL, showing that TGF-β1 treatment enhanced cell migration. In contrast, PTL inhibited TGF-β1-induced EMT, reducing CRC cell migration and invasion, which is crucial for preventing metastasis. Additionally, the TGF-β1 treatment of SW480 and HT-29 cells resulted in a decrease in E-cadherin levels and an increase in vimentin, β-catenin, Snail, and Slug expression. However, when PTL was administered to TGF-β1-treated SW480 and HT-29 cells, vimentin, β-catenin, Snail, and Slug levels decreased, while E-cadherin expression increased [132].

### 3.19. Scabertopin

Scabertopin is a type of SL that can be extracted from *Elephantopus scaber* L. This SL has the ability to suppress the growth of tumor cells [133]. In an in vitro study on bladder cancer using J82 cell lines, treatment with scabertopin for 24 h was evaluated using transwell and wound healing assays. The results demonstrated a significant reduction in cell migration and invasion. Furthermore, since the activation of the FAK/PI3K/Akt signaling axis is critical for cancer cell invasiveness, Western blot analysis revealed that scabertopin treatment led to a marked decline in the expression of p-PI3K, PI3K, p-AKT, AKT, p-FAK, FAK, and MMP-9 [134].

## 4. Concluding Remarks and Future Directions

Cancer is a multifactorial disease, necessitating the development of diverse anti-cancer therapies. Focusing on a single aspect, such as the cell death mechanisms or metabolism, is no longer sufficient. A multitarget approach is essential to address the complexities of cancer biology, utilizing a combination of natural and synthetic agents that synergistically target multiple signaling pathways involved in carcinogenesis. According to a recent report, at least one-third of all drugs have been derived from plant materials [135]. Recently, phytochemicals have shown promising potential as single effective compounds or adjuvants to conventional anti-cancer therapies.

As metastasis is an important mechanism by which cancer cells spread to other organs and develop new and aggressive forms of the primary tumor, it is essential to understand this mechanism and seek new ways to prevent this detrimental process in cancers. The focus of this present review is on the anti-invasion, -migration, and -metastatic aspects of sesquiterpene lactones in various cancer types. As shown in Table 1, these phytochemicals act on multiple molecular targets in the metastatic cascade of cancer cells. This shows that sesquiterpene lactones are promising anti-metastatic drug candidates for the development of new drugs with different targets in a variety of cancer types. This review reported that 13 articles have investigated the effects of sesquiterpene lactones on EMT markers. As described earlier, EMT is the initial step in metastasis, allowing tumor cells to gain migratory and invasive properties. Therefore, targeting this process is a critical step for the suppression of metastasis of tumors. This implies that future studies should conduct novel methods to evaluate the effect of sesquiterpene lactones on the EMT process to prevent tumor cell metastasis. Angiogenesis is another process that is very crucial for cancer cells to be transported to other organs. Our literature review showed that only nine studies have examined the effects of sesquiterpene lactones on angiogenesis or related factors in cancer models. This indicates the lack of enough data to establish the anti-metastatic effects of some sesquiterpene lactones and shows that there is also room for further investigations.

This review identified only 18 studies examining the anti-metastatic effects of sesquiterpene lactones in animal models. This limited number of in vivo studies poses a challenge in translating the findings to effective doses in human subjects. Although only about 8% of outcomes from animal studies successfully translate to clinical trials, these models provide valuable in vivo insights into the effects of cancer therapies [136]. Due to ethical and practical concerns with human studies, the development of suitable animal models is pivotal in cancer research. To the best of our knowledge, no previous clinical trial has investigated the effect of these phytochemical compounds on metastatic cancers in humans, and the literature lacks essential data in this regard. Therefore, the current studies fail to address the effectiveness of these compounds in clinical settings. The lack of essential data not only hinders our understanding of the efficiency of sesquiterpene lactones on cancer metastasis but also hampers the development of effective drug candidates from these phytochemicals for suppressing cancer metastasis.

Bioavailability represents a significant challenge associated with therapeutic agents derived from natural sources [137]. Therefore, it is imperative to carry out pharmacokinetic studies to evaluate the bioavailability of therapeutic phytochemicals in order to identify the most suitable candidate for subsequent therapeutic assessment. A particularly promising and effective strategy for enhancing the bioavailability and targeting of pharmaceuticals, especially natural compounds, is the utilization of nanomaterial structures [138].

## Figures and Tables

**Figure 1 biomolecules-15-00268-f001:**
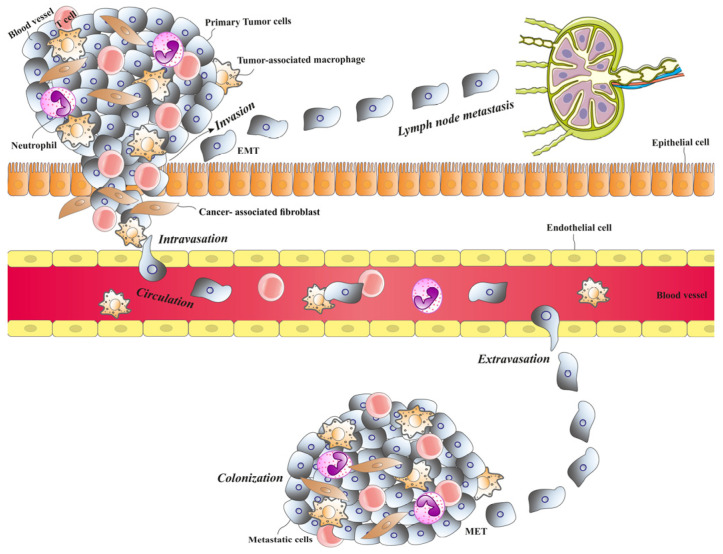
Cancer cell metastasis involves several steps. Primary tumor cells invade nearby tissue, then intravasate into blood vessels or lymph nodes, allowing transport to other organs. Once in the bloodstream, circulating tumor cells extravasate into a secondary organ and begin to proliferate, forming a metastatic colony.

**Figure 2 biomolecules-15-00268-f002:**
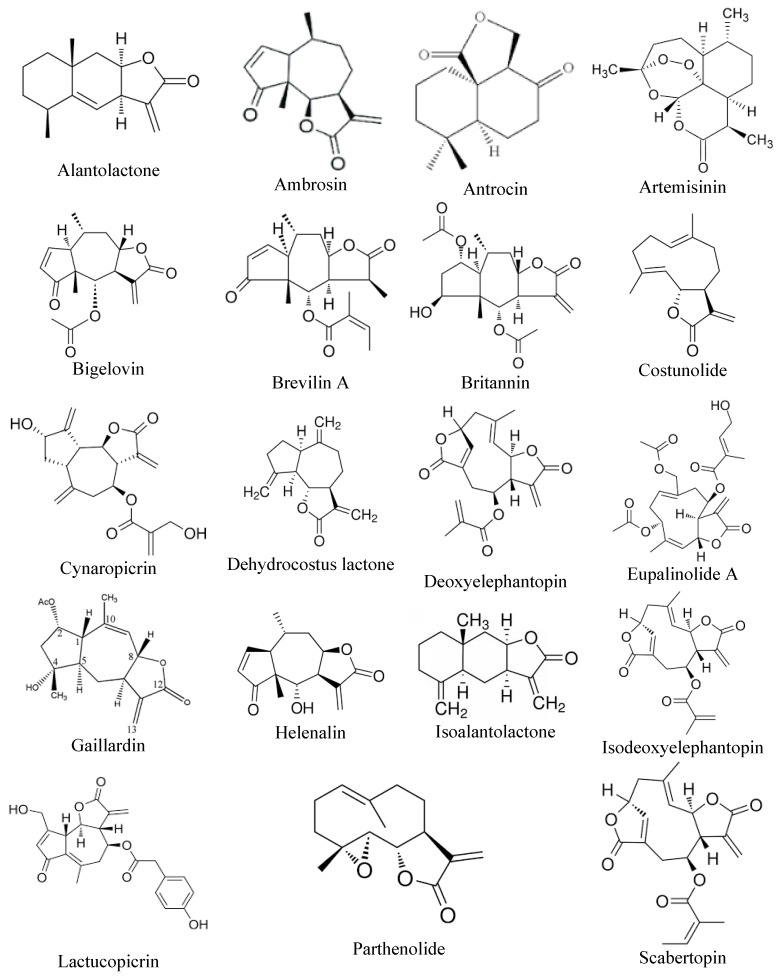
Chemical structures of sesquiterpene lactones that have anti-metastatic activity.

**Table 1 biomolecules-15-00268-t001:** Sesquiterpene lactones with anti-metastatic effects on different cancers.

Compound	Plant Source	Concentration	Cancer Type	Cancer Model	Altered Factors	Ref.
Alantolactone	*Inula helenium*	1–100 μM for cell lines	Breast	HUVEC cells for CAM assay	Reduced cell mobility and migration; suppressed new blood vessel formation	[46]
		5 mg/kg/day over 15 days for animal model		BABL/c nude mice with MDA-MB-231 xenografts		
		10, 20, and 30 μM	Breast	MCF-7 cells	Suppressed colony formation and migration; inhibited invasion and cell migration by downregulating MMP-2, MMP-7, and MMP-9	[47]
		2.5, 5, and 10 μM for cell lines	Melanoma	A375, A2058, A375R (vemurafenib-resistant variant of A375), and HK-2 cell line	Inhibited STAT3 signaling	[48]
		10 mg/kg over 12 days for animal model		BALB/c nude mice		
		0.01 and 0.1 μM	Prostate	PC3 cells	Antagonized the STAT3 signaling pathway, leading to the upregulation of p53 and the downregulation of Oct-4, CD44, CD133, and Nanog expression, reduced stemness traits, and inhibited migration in metastatic prostate cancer cells	[49]
		U2OS, 0, 5, 10, or 20 μM; HOS, 0, 15, 30, or 60 μM	Osteosarcoma	U2OS and HOS cells	Reduced cell migration, invasion, and aggressiveness	[50]
		0, 4, 8, and 10 μM for cell lines	Osteosarcoma	143B, MG63, U2OS, and SaoS2	Decreased EMT-related markers, including vimentin, Snail, and N-cadherin; increased the epithelial marker E-cadherin; reduced cell invasion, migration, and proliferation; decreased in the expression of MMP-9, MMP-2, and MMP-7	[51]
		5, 15, and 25 mg/kg over 21 days for animal model		Athymic mice		
		3, 10, and 30 μM for cell lines	Lung	NCI-H460 cell lines	Inhibited AKR1C1, resulting in a reduction in AKR1C1 expression; decreased metastasis and cell growth	[52]
		10 and 20 mg/kg over 21 days for animal model		BALB/c nude mice		
Ambrosin	*Hymenoclea salsola* and *Ambrosia maritima*	1, 2.5, or 5 μM	Breast	MCF-7, JIMT-1, and HCC1937, MCF-10A	Reduced the populations of breast cancer stem cells; inhibited cell migration	[55]
		8, 32, and 64 μM	Breast	MDA-MB 231 cells	Inhibited cell proliferation; downregulation of the Wnt/β-catenin pathway	[56]
		12.5, 25, and 50 μM	Breast	MDA-MB 231	Inhibited colony formation; reduced MMP expression; decreased the levels of phosphorylated GSK-3β and Akt, thereby inhibiting the Akt/β-catenin signaling pathway	[57]
Antrocin	*Antrodia camphorata*	25, 50, and 75 μg/mL	Bladder	5637 and T24	Inhibited invasion, migration, and cell proliferation; reduced the phosphorylation of FAK and paxillin, leading to the disruption of filopodia and lamellipodia formation; increased E-cadherin levels; decreased vimentin expression; reduced MMP-2 activity	[59]
		50, 100, and 200 μM for cell lines	Breast	MCF7 and MDA-MB-231	Downregulated the expression of oncogenes and stemness-related markers such as β-catenin, Akt, and Notch1; reduction in migration, tumorigenesis, and proliferation	[60]
		30 mg/kg over two weeks for animal model		NOD/SCID mice		
		20, 50, 100, 200, and 300 µM	Kidney	RCC 786-0 cells	Inhibited the Src, FAK, and ERK1/2 signaling pathways, leading to a reduction in the phosphorylation of paxillin, C/EBP-β, and total c-Fos levels; decreased the expression of MMP-7 and vimentin; disrupted cell migration, invasion, and the formation of lamellipodia	[61]
Artemisinin	*Artemisia annua*	7.5, 15, or 30 μM	Lung	A549 and H1299 cell lines	Inhibited migration and invasion; suppressed the activity of MMPs; suppressed the expression of EMT-related proteins, including N-cadherin and vimentin, and cancer stem cell (CSC) markers like Nanog, Sox2, and Oct3/4; increased E-cadherin expression; disrupted the Wnt/β-catenin signaling pathway	[64]
		12.5, 25, 50, or 75 μM	Liver	HepG2 and SMMC-7721 cells	Decreased the expression of MMP2 and the upregulation of TIMP2; inhibited the activation of p38 and ERK1/2; enhanced cell adhesion by increasing Cdc42 activity, which activated E-cadherin; inhibited cell motility, migration, and metastasis	[65]
		2.5 μM	Lung	H1395, A549, LXF289 cells, Calu3 and H1299, H460	Inhibited the expression of MMP-2, MMP-7, and u-PA, leading to the inhibition of metastasis and invasion	[66]
Brevilin A	*Centipeda minima*	0.25 and 0.5 μM for cell lines	Melanoma	A375 and A2058	Inhibited the JAK2/STAT3 pathway by reducing the phosphorylation of JAK2 and STAT3; suppressed cell invasion and migration	[69]
		4.5 mg/kg and 9 mg/kg over 21 for animal model		nu/nu BALB/c mice		
		5, 10, and 15 μM	Liver	HepG2 and SMMC-7221	The downregulation of MMP-2 and MMP-9 inhibited the Wnt/β-catenin and STAT3/Snail signaling pathways, resulting in decreased cell invasion	[90]
		1.25, 2.5, 5, 10, 20 μM for cell lines	Breast	MDA-MB 231 and MDA-MB 468	Inhibited cell migration and reduced the phosphorylation and expression of Akt, mTOR, and STAT3, thereby suppressing the Akt-mTOR and STAT3 signaling pathways	[71]
		25 and 50 mg/kg/day over 22 days for animal model		BALB/c nude mice		
		2.5, 5, and 10 μM	Colorectal	HCT-116 and CT26	Suppressed the expression of MMP-2 and VEGF; inhibited STAT3; suppressed angiogenesis; reduced cell migration and invasion	[72]
		2.5, 5, and 10 μM for cell lines	Colorectal	LOVO, HCT-116, HT29, and CT26, NCM460, human hepatic stellate cell line LX-2, and the mouse hepatic stellate cell line JS1	Inhibited colorectal liver metastasis and tumor growth by targeting the VEGF-IL6-STAT3 axis	[73]
		4 and 8 mg/kg over 2 weeks for animal model				
				BALB/c mice		
Bigelovin	*Inula helianthus-aquatica* C. Y. Wu	Zebrafish embryos: 25, 50, and 100 μM; endothelial cells (HMEC-1): 400–1600 nM; human PBMCs: 62.5–250 nM; monocyte adhesion (THP-1 to HMEC-1): 500–4000 nM	Non-cancerous cell lines	Zebrafish embryos, endothelial cells (HMEC-1), human PBMCs, and THP-1 monocytes	Suppressed the formation of subintestinal vessels in zebrafish embryos; induced anti-angiogenic effects by downregulating angiogenesis-related genes (Ang-1, Ang-2, Tie-1, and Tie-2), reducing Th1 cytokine production (IFN-γ, IL-2, and IL-12), and inhibiting CAM gene expression (ICAM-1, VCAM-1, and E-selectin)	[75]
		0.75, 1.5, or 3 μM for cell lines	Colon	Colon 26-M01, and HCT116	Significant changes in key molecules, including p-STAT3, STAT3, Rock, β-catenin, N-cadherin, Rac1/2/3, and RhoA, resulting in the disruption of the IL6-STAT3 and cofilin pathways; inhibited cell motility, migration, EMT, angiogenesis, and cell growth; suppressed liver and lung metastasis	[76]
		0.3, 1, and 3 mg/kg over 18 days for animal model		BALB/c mice		
Britannin	*Inula britannica* L.	20, 40, and 80 μM	Gastric	AGS and MKN-45	Reduction in the expression of MMP-9, TWIST-1, and COX-2	[78]
		2.7 and 6 μM	Liver	BEL-7402 and HepG2	Suppressed tumor cell migration	[79]
		1, 3, and 10 μM for cell lines	Colorectal, lung, cervical, and liver	HCT116, A549, HeLa, Hep3B, HUVECs	Inhibited invasion, migration, and angiogenesis by reducing PD-L1 levels; suppressed the expression of VEGF and MMP-9	[80]
				BALB/c nude mice		
		5 mg/kg or 15 mg/kg over 30 days for animal model				
		5, 10, and 20 μM	Lung	A549	Reduced KLF5 expression; inhibited cell migration	[81]
Costunolide	*Saussurea lappa*	20 and 50 μM for cell lines	Breast	MDA-MB 231	Inhibited NF-*κ*B, which led to the suppression of TNF*α*-induced migration and invasion of cancer cells; downregulation of MMP-9; suppressed metastasis	[83]
		20 μM over 30 days for animal model		Nude (Nu/Nu) mice		
		2.5, 5, and 10 μM	Colorectal	HCT-15, HCT-116, and DLD1	Reduction in cell migration and invasion; decreased vimentin and N-cadherin levels; increased E-cadherin expression	[84]
		1, 5, and 25 μM for cell lines	Skin	HUVECs, human epidermoid carcinoma KB3-1 cells	Interfered with the VEGFR KDR/Flk-1 signaling pathways related to angiogenic factors, leading to the suppression of pro-angiogenic activity	[85]
		100 mg/kg over 7 days for animal model		BALB/c mice		
		12, 24, and 48 μM	Lung	H1299	Inhibited cell migration and invasion, suppressed the EMT process by the upregulation of E-cadherin and a downregulation of N-cadherin; reduction in mRNA expression levels of integrins α2 and β1, as well as the MMP2	[86]
		0.1–10 μM	Neuroblastoma	NB-39	inhibition of cell migration and invasion, and the downregulation of MMP-2	[87]
Cynaropicrin	*Cynara scolymus* L.	5, 7.5, 10 μM	Colorectal	HCT116, RKO, and DLD-1	Reduction in cell migration	[89]
		3, 10, 30 μM	Melanoma	A375	Reduction in the MAPK/ERK and NF-κB pathways; inhibited cell motility and invasion	[90]
Dehydrocostus lactone	*Aucklandiae Radix*	2.0, 4.0, 8.0, and 16.0 μM	Lung	H1299 cells	Inhibited cell migration and invasion; upregulation of the expression of E-cadherin; downregulation of N-cadherin, Snail, integrin α2, and MMP-2	[91]
		3 μg/mL	Larynx	TU212 and HBE	Downregulation of MMP-2 and MMP-9; inhibited cell migration and invasion	[92]
Deoxyelephantopin	*Elephantopus scaber*	HCT 116 (3.73), K562 (0.5), KB (0.41), and T47D (0.91) μg/mL	Colorectal, chronic myeloid leukemia, oral, and breast	HCT 116, K562, KB, and T47D cells	Suppressed cell migration and invasion; decreased the expression of uPA, uPAR, MMP-2, and MMP-9; upregulation of TIMP-1 and TIMP-2	[94]
		12.28 µg/mL	Lung	A549 cells	Decreased migration and invasion; decreased the expression levels of NF-kB, IkBa, MMP-2, MMP-9, uPA, and uPAR; increased TIMP-2 levels; reduced protein levels of p-ERK 1/2 and p-Akt, along with increased levels of p-p38 and p-JNK, contributing to metastasis suppression	[95]
		1.5, 3, and 6 μM for cell lines	Melanoma	A375LM5^IF4g/Luc^	Inhibited N-cadherin, MMP2, vimentin, and integrin-4; suppressed pulmonary vascular permeability, VEGF+, neovascularization marker CD31, and N-cadherin	[96]
		20 mg/kg over 27 days for animal model		NOD/SCID mice		
		1.0, 2.5, 5.0, and 150 μM	Adenocarcinoma	TS/A cells	Inhibited cell motility; disrupted adhesion formation by inhibiting m-calpain’s enzymatic activity; inhibited lamellipodia formation and actin filament organization	[97]
		0.5 to 3 μM for cell lines	Melanoma	B16 murine melanoma cell line	Inhibited cell migration, suppressed lung metastasis	[98]
		10 mg/kg over 21 days for animal model		C57BL/6J mice		
Eupalinolide	*Eupatorium lindleyanum* DC.	10 μM	Breast	MDA-MB-468, MDA-MB231	STAT3 inhibitory effects	[101]
		2.5, 5, and 10 μM	Pancreas	MiaPaCa-2, PANC-1, and PL-45	Inhibition of cell migration and invasion	[102]
		4 and 8 μM	Breast	MDA-MB-231, MDA-MB-468, and MCF-7	Downregulation of STAT3 and p-STAT3	[100]
		1.25 and 2.5 μM for cell lines	Breast	U251 and MDA-MB-231 cells	Decreased in STAT3, p-STAT3, MMP-9, and MMP-2; suppressed cancer cell metastasis	[103]
		30 mg/kg over 18 days for animal model		BALB/c nu/nu mice		
		7, 14, or 28 μM	Liver	MHCC97-L and HCCLM3	Suppressed cell motility and migration through the downregulation of vimentin and the upregulation of ZEB1, N-cadherin, and fibronectin	[104]
		1 and 2 μM	Larynx	TU212 cells	Inhibited EMT markers by a decrease in N-cadherin expression and an increase in E-cadherin; reduction in cell motility; suppressed LSD1	[105]
Gaillardin	*Inula oculus-christi*	20, 40, and 80 μM	Gastric	MKN45 and AGS	Suppressed NF-κB, which subsequently downregulated its target genes, including COX-2, TWIST-1, and MMP-9	[107]
Helenalin	*Arnica montana* and *Arnica chamissonis ssp. Foliosa*	2.5 and 5 μM	Rhabdomyosarcoma	RH30 and RD cells	Inhibited cell migration	[109]
Isoalantolactone	*Inula helenium* L.	25, 75, and 150 μM	Liver	Hep-G2	Reduced cell invasion and migration	[111]
		1, 2, and 4 μM	Breast	MDA-MB-231	Inhibited the p38 and MAPK/NF-κB signaling pathways, leading to the suppression of cell migration and invasive activities	[112]
		2 or 4 μg/mL	Pancreas	PANC-1 and SW1990	Suppressed colony formation and cell migration	[110]
		5, 10, and 20 μM	Colorectal	HCT116 and SW620	Suppressed colony formation	[113]
		5, 10, and 20 μM	Endometrium	HEC-1-B	Inhibited both migratory properties and invasiveness	[114]
		4.5, 9.0, or 18 μM	Liver	HuH7	Inhibition of cell invasion	[115]
		20 μM	Pancreas	PANC-1, AsPC-1, and BxPC-3	Suppressed cell migration and invasion	[116]
Isodeoxyelephantopin	*Elephantopus scaber* L.	0.4, 0.8, 1.6, 3.2, 6.4, and 12.8 μM	Lung	H1299 and A549	Suppressed colony formation	[118]
		50 μM	Breast	MDA-MB-231	Suppressed cell migration; inhibited the expression of MMP-2 and MMP-9	[119]
		2 μM	Lung	H1299	Inhibited cell invasion; suppressed the expression of MMP-9 and ICAM-1	[120]
Lactucopicrin	*Lactuca virosa*	7.5, 15, and 30 μM	Skin	SKMEL-5	Reduction in p-PI3K, p-Akt, and p-mTOR levels	[122]
		12.5, 25, and 50 μM	Osteosarcoma	Saos-2 cells	Inhibited cell migration and invasion	[123]
		7.5 and 10 μM	Glioblastoma	U87MG cells	Reduction in colony formation; decreased cell motility; reduction in Akt phosphorylation levels	[124]
Parthenolide	*Tanacetum parthenium*	4 and 8 μM	Kidney	786-O and ACHN cells	Decreased cell proliferation; inhibited both cell migration and invasion; suppressed MMP-2 and MMP-9 expression levels; increased E-cadherin levels; decreased N-cadherin, vimentin, and Snail levels; inhibited ALDH1, CD133, Oct4, and Sox2; reduction in the number of spheres; decline in p-PI3K and p-AKT expression	[125]
		2 μM		MCF-7	Downregulated vimentin expression	[127]
		2.5, 5, 10, 20, and 40 μM for cell lines	Colorectal	HT-29, DLD-1, and HCT116 cells	Decreased HIF-1α expression, suppressed migration and invasion; reduced the expression of MMP-2 and MMP-9; increased E-cadherin levels; downregulated EMT markers such as β-catenin, vimentin, Slug, Snail, and Twist; reduced levels of CA IX, a marker of hypoxia; decreased the number of NF-κB subunit p65-positive cells; reduced angiogenesis marker von Willebrand factor (VWF) and EMT marker vimentin	[128]
		4 mg/kg over 27 days for animal model				
				Female athymic nude mice		
		5, 10, and 20 μM	Colorectal	SW620 cells	Inhibited cell migration and invasion; upregulation of E-cadherin; downregulation of β-catenin, Snail, vimentin, MMP-2, MMP-9, and COX-2 expression	[129]
		0.2, 2, 10, 20, and 200 μg/mL for cell lines	Osteosarcoma	LM8 cells	Suppression of NF-κB DNA binding and transcriptional activity; reduction in VEGF expression; inhibited tumor invasion; reduced pulmonary metastasis; decreased p65 expression; suppressed VEGF expression in metastatic lung tumors and surrounding lung tissue	[130]
		0.01, 0.1, 1, or 100 μg/kg or 1 mg/kg daily over 25 days for animal model		C3H male mice		
		15, 25, and 50 μM for cell lines		MDA-MB 231	Downregulation of NF-κB activity; inhibited cell migration; reduced vimentin expression; reduction in the levels of VEGF, MMP2, MMP9, and p65	[131]
		50 mg/kg daily over 16 days for animal model		Nude athymic mice		
		5 μM	Colorectal	SW480 and HT-29 cells	Inhibited TGF-β1-induced EMT; reducing cell migration and invasion, decreased vimentin, β-catenin, Snail, and Slug levels; increased E-cadherin expression	[132]
Scabertopin	*Elephantopus scaber* L.	10 and 15 μM	Bladder	J82 cells	Suppressed cell migration and invasion; decreased the expression levels of MMP-9, phospho-FAK (Tyr397), phospho-AKT (Ser472, Ser473, Ser474), and phospho-PI3K (Tyr607)	[134]

## Data Availability

No new data were created or analyzed in this study.

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
