# Peer review of "Sesquiterpene Lactones as Promising Phytochemicals to Cease Metastatic Propagation of Cancer"

_biomolecules, 2025, doi:10.3390/biom15020268_

Round 1
Reviewer 1 Report
Comments and Suggestions for Authors
General comments to the paper entitled: Sesquiterpene lactones as promising phytochemicals to cease metastatic propagation of cancer
The authors aim to summarize our knowledge about how cancer cells spread in the body and show the role of phytochemicals (sesquiterpene lactones) in inhibiting invasion, migration, and metastasis growth.
The authors adequately summarize the key steps of metastasis: invasion, angiogenesis, intravasation, circulation, and colonization.
The paper introduces 19 different molecules belonging to the sesquiterpene lactones group and discusses very accurately one by one the available results obtained in vitro and in vivo experiments. Table 1 precisely summarizes all the details of the related experiments. All the results come from in vitro and in vivo experiments.
I suggest that discussing one sesquiterpene lactone would be useful for the reader to know which plant contains the given molecules.
The reader can see the complexity of tumor development and the mechanism of how the tumor spreads in the body. The authors emphasize that they cannot cite human clinical data to confirm the effect of sesquiterpenes inhibiting the appearance of tumor metastasis. This is understandable considering the cost and ethical difficulties of conducting a clinical trial.
It would be interesting to find some correlation between the presence of a given plant in the population's diet and the reduced incidence of cancer compared to other regions.
Fig. 1. The same kind of cells are indicated as blood vessels and primary tumors. I suggest removing the blood vessel from the top left of the figure.
Author Response
Marseille, the 03th Feb 2025
Dear Editor, Dear Reviewer,
We sincerely thank the reviewer for constructive criticism and valuable comments, which were very helpful in revising the manuscript. Accordingly, the revised manuscript has been systematically improved with new information and additional interpretations. Our responses (AR) to the referee’s comments are given below. In addition, We have highlighted all the changes in the manuscript in yellow color.
Reviewer 1 :
General comments to the paper entitled: Sesquiterpene lactones as promising phytochemicals to cease metastatic propagation of cancer. The authors aim to summarize our knowledge about how cancer cells spread in the body and show the role of phytochemicals (sesquiterpene lactones) in inhibiting invasion, migration, and metastasis growth. The authors adequately summarize the key steps of metastasis: invasion, angiogenesis, intravasation, circulation, and colonization. The paper introduces 19 different molecules belonging to the sesquiterpene lactones group and discusses very accurately one by one the available results obtained in vitro and in vivo experiments. Table 1 precisely summarizes all the details of the related experiments. All the results come from in vitro and in vivo experiments.
(AR) We thank the reviewer for this nice comment and let us know about the strengths of our work.
I suggest that discussing one sesquiterpene lactone would be useful for the reader to know which plant contains the given molecules.
(AR) We appreciate the reviewer’s perspective. We believe that we discussed all of them in the manuscript to help the reader know the plant-derived SLs.
The reader can see the complexity of tumor development and the mechanism of how the tumor spreads in the body. The authors emphasize that they cannot cite human clinical data to confirm the effect of sesquiterpenes inhibiting the appearance of tumor metastasis. This is understandable considering the cost and ethical difficulties of conducting a clinical trial.
(AR) Thanks for the clarification.
It would be interesting to find some correlation between the presence of a given plant in the population's diet and the reduced incidence of cancer compared to other regions.
(AR) We appreciate the reviewer’s perspective. We agree that finding these correlations would provide useful and important data, but the literature lacks these types of studies to assess the correlation between the presence of a given plant in the population's diet and the reduced incidence of cancer. To the best of our knowledge, no previous study has evaluated this correlation regarding the plant materials explained in our review paper.
Fig. 1. The same kind of cells are indicated as blood vessels and primary tumors. I suggest removing the blood vessel from the top left of the figure.
(AR) In accordance with the reviewer’s suggestion, we have removed the blood vessels on the figure.
Regards
Dr M Maresca
Reviewer 2 Report
Comments and Suggestions for Authors
This is a comprehensive review article on the role of sesquiterpene lactones in inhibiting cancer metastasis. The review article delves deeper into the mechanisms by which these compounds exert their anti-metastatic effects on different types of cancer. It introduces sesquiterpene lactones, plant-derived compounds with potent anti-cancer properties. It emphasizes their ability to target and inhibit key processes in metastasis, such as invasion, migration, and the spread of cancer cells. This review also provides a thorough overview of the metastatic process, including key steps like invasion, intravasation, extravasation, and colonization. Although this review has its strengths in introducing seq. lactones as anti-cancer agents, but this is not a novel topic to introduce. It has been cited and reviewed at lengths and breadths. The reviewers have tried to condense all the properties of this phytochemical together which is promising to readers. However, this lacks a thorough and precise accumulation of the findings. The article is poorly written and have a lot of redundant information as well as citations. Please address the following concerns to make it interesting for readers:
1. References:
a. Some citations are redundant at a few places such as EMT 13, 24, 25, hallmarks of metastasis – please exclude the ones which say the same thing repeatedly. Include precise citations. #26-29: do not overcite references for the same thing.
b. 32 and 34 are basically cited for the same thing. #34 and 40?? #35 and 50?
c. Cited review#11 only explains matcha tea. Please include a citation with broad spectrum of phytochemicals. Review# 10 and 12 are relevant here not 11.
d. How relevant is the cited review #55? Please remove it.
e. #56 is again recited for already cited review. Please mention the previously cited review for metastases
f. #1 and #57 are same citation.
g. Please explain how the Fig. 2 is relevant when you have only shown the structure but haven’t discussed.
h. What’s the relevance of #63?
i. How are #65 and 69 different when you recite either of them to explain STAT3 signaling.
j. Correct typos wherever required such as : Rashee et al. line 288
k. Section 3.8: why have you not cited review#60 and added more. Please avoid that whwrever required.
2. Metastasis as a hallmark of cancer and its consequences: Lines 32-35: These lines repeatedly emphasize that metastasis is a critical hallmark of cancer and a major cause of cancer-related deaths. This information is sufficiently covered in lines 32 and 33.
3. Definition and stages of metastasis: Lines 37-83: This section extensively describes the process of metastasis, including its stages (invasion, intravasation, etc.). While important, some of the details, particularly in lines 71-83, could be condensed or summarized to avoid repetition.
4. EMT as a key driver of metastasis: Lines 49-52 and lines 90-106: Both sections describe the epithelial-mesenchymal transition (EMT) and its role in enabling cancer cell invasion. This information is redundant and could be presented more concisely.
5. Angiogenesis as a crucial step for tumor growth: Lines 108-119: This section describes angiogenesis and its importance for tumor growth. While relevant, some of the information might be considered background knowledge and could be summarized or presented more concisely.
6. Role of the tumor microenvironment: Lines 42-48: This section discusses the role of the tumor microenvironment in metastasis. Some of the information, particularly the general description of the TME components, could be condensed.
7. Please Condense repetitive information: Combine or remove redundant sentences or paragraphs to improve the flow and conciseness of the text.
8. Streamline descriptions: Summarize or condense detailed descriptions of well-established concepts like metastasis stages and the EMT process.
9. Focus on novel findings: Emphasize the unique contributions of the cited studies, such as specific mechanisms of action of sesquiterpene lactones, rather than repeating general knowledge about metastasis.
10. Please also highlight then limitations of phytochemical to be used as therapeutic agents with a challenge in bioavailability or whatever is relevant in this context.
Comments on the Quality of English Language
There are a few typos and English language could be improved to make the flow readable.
Author Response
Marseille, the 03th Feb 2025
Dear Editor, Dear Reviewer,
We sincerely thank the reviewer for constructive criticism and valuable comments, which were very helpful in revising the manuscript. Accordingly, the revised manuscript has been systematically improved with new information and additional interpretations. Our responses (AR) to the referee’s comments are given below. In addition, We have highlighted all the changes in the manuscript in yellow color.
Reviewer 2:
This is a comprehensive review article on the role of sesquiterpene lactones in inhibiting cancer metastasis. The review article delves deeper into the mechanisms by which these compounds exert their anti-metastatic effects on different types of cancer. It introduces sesquiterpene lactones, plant-derived compounds with potent anti-cancer properties. It emphasizes their ability to target and inhibit key processes in metastasis, such as invasion, migration, and the spread of cancer cells. This review also provides a thorough overview of the metastatic process, including key steps like invasion, intravasation, extravasation, and colonization. Although this review has its strengths in introducing seq. lactones as anti-cancer agents, but this is not a novel topic to introduce. It has been cited and reviewed at lengths and breadths. The reviewers have tried to condense all the properties of this phytochemical together which is promising to readers. However, this lacks a thorough and precise accumulation of the findings. The article is poorly written and have a lot of redundant information as well as citations. Please address the following concerns to make it interesting for readers:
(AR) We appreciate the reviewer highlighting the strengths of our work alongside its limitations. It is important to clarify that our topic is novel, as no prior review has specifically addressed the anti-metastatic properties of sesquiterpene lactones across various cancers, nor has any paper summarized their mechanisms of action in inhibiting cancer metastasis. Thus, the novelty of our work is well-established. The reviewer’s comment about a lack of thoroughness is unclear to us, as we have compiled and clearly presented all available studies on the anti-cancer effects of sesquiterpene lactones and their mechanisms. We believe we have covered all relevant findings in the literature. Additionally, we acknowledge the reviewer’s feedback regarding redundant references and information, and we have revised these accordingly.
- References:
- Some citations are redundant at a few places such as EMT 13, 24, 25, hallmarks of metastasis – please exclude the ones which say the same thing repeatedly. Include precise citations. #26-29: do not overcite references for the same thing.
(AR) According to the reviewer’s suggestion, we have removed references 24, 25, 27, and 29. It is noteworthy that ref 26 is now 24 in the text.
- 32 and 34 are basically cited for the same thing. #34 and 40. #35 and 50.
(AR) According to the reviewer’s suggestion, we have removed references 34 and 50.
- Cited review#11 only explains matcha tea. Please include a citation with broad spectrum of phytochemicals. Review# 10 and 12 are relevant here not 11.
(AR) According to the reviewer’s suggestion, we have removed reference 11 and replaced it with reference 12.
- How relevant is the cited review #55? Please remove it.
(AR) According to the reviewer’s suggestion, we have removed reference 55 and the corresponding sentence.
- #56 is again recited for already cited review. Please mention the previously cited review for metastases.
(AR) According to the reviewer’s suggestion, we have removed reference 56 and the corresponding sentence.
- #1 and #57 are same citation.
(AR) According to the reviewer’s suggestion, we have removed reference 57 and replaced it with reference 1.
- Please explain how the Fig. 2 is relevant when you have only shown the structure but haven’t discussed.
(AR) We appreciate the reviewer's perspective, but we believe that it is crucial to include the chemical structures of phytochemicals in this article. This addition aids readers in visualizing the different structures and understanding the relationship between such big structures and their anticancer activities. However, if the reviewer prefers to omit them, we can accommodate that request.
- What’s the relevance of #63?
(AR) According to the reviewer’s suggestion, we have removed reference 63.
- How are #65 and 69 different when you recite either of them to explain STAT3 signaling.
(AR) According to the reviewer’s suggestion, we have removed reference 65. But reference 69, which now is 56, explains that ALT can reduce stemness traits and inhibit migration in metastatic prostate cancer cells. So, we didn't remove it.
- Correct typos wherever required such as : Rashee et al. line 288.
(AR) We thank the reviewer for pointing out this error. We have revised it.
- Section 3.8: why have you not cited review#60 and added more. Please avoid that whwrever required.
(AR) The reference 60 is about the effects of britannin on apoptosis and autophagy, not metastasis and the present review article specifically focuses on metastasis.
- Metastasis as a hallmark of cancer and its consequences: Lines 32-35: These lines repeatedly emphasize that metastasis is a critical hallmark of cancer and a major cause of cancer-related deaths. This information is sufficiently covered in lines 32 and 33.
(AR) We thank the reviewer for this suggestion. We have summarized this section according to the reviewer's comments.
- Definition and stages of metastasis: Lines 37-83: This section extensively describes the process of metastasis, including its stages (invasion, intravasation, etc.). While important, some of the details, particularly in lines 71-83, could be condensed or summarized to avoid repetition.
(AR) We thank the reviewer for this excellent suggestion. We have summarized this section according to the reviewer's comments.
- EMT as a key driver of metastasis: Lines 49-52 and lines 90-106: Both sections describe the epithelial-mesenchymal transition (EMT) and its role in enabling cancer cell invasion. This information is redundant and could be presented more concisely.
(AR) We thank the reviewer for pointing out this error. We have removed the redundant section in the introduction (Lines 49-52).
- Angiogenesis as a crucial step for tumor growth: Lines 108-119: This section describes angiogenesis and its importance for tumor growth. While relevant, some of the information might be considered background knowledge and could be summarized or presented more concisely.
(AR) According to the reviewer’s suggestion, we have summarized this section carefully.
- Role of the tumor microenvironment: Lines 42-48: This section discusses the role of the tumor microenvironment in metastasis. Some of the information, particularly the general description of the TME components, could be condensed.
(AR) According to the reviewer’s suggestion, we have summarized this section.
- Please Condense repetitive information: Combine or remove redundant sentences or paragraphs to improve the flow and conciseness of the text.
(AR) According to the reviewer’s suggestion, we have summarized the suggested sections to avoid redundancy.
- Streamline descriptions: Summarize or condense detailed descriptions of well-established concepts like metastasis stages and the EMT process.
(AR) According to the reviewer’s suggestion, we have summarized the detailed descriptions of well-established concepts.
- Focus on novel findings: Emphasize the unique contributions of the cited studies, such as specific mechanisms of action of sesquiterpene lactones, rather than repeating general knowledge about metastasis.
(AR) With all due respect to the reviewer, we should explain that we have tried to explain the novel aspects of the studies by ignoring or removing other unrelated sections. Besides, we have focused on different mechanisms by which the sesquiterpene lactones suppress metastatic cascade or angiogenic switch in a variety of cancer cells. However, if the reviewer would like to add any specific data, we are ready to do them carefully.
- Please also highlight then limitations of phytochemical to be used as therapeutic agents with a challenge in bioavailability or whatever is relevant in this context.
(AR) We thank the reviewer for this suggestion. We have addressed this issue and the potential solution in the conclusion section.
Regards
Dr M Maresca